# Multilineage Differentiation for Formation of Innervated Skeletal Muscle Fibers from Healthy and Diseased Human Pluripotent Stem Cells

**DOI:** 10.3390/cells9061531

**Published:** 2020-06-23

**Authors:** Kilian Mazaleyrat, Cherif Badja, Natacha Broucqsault, Raphaël Chevalier, Camille Laberthonnière, Camille Dion, Lyla Baldasseroni, Claire El-Yazidi, Morgane Thomas, Richard Bachelier, Alexandre Altié, Karine Nguyen, Nicolas Lévy, Jérôme D. Robin, Frédérique Magdinier

**Affiliations:** 1Aix-Marseille University, INSERM, MMG, Marseille Medical Genetics, 13385 Marseille, France; mazaleyrat.kilian@gmail.com (K.M.); cherifbadja@hotmail.fr (C.B.); natacha.broucqsault@univ-amu.fr (N.B.); raphael.chevalier@univ-amu.fr (R.C.); camllle.laberthonniere@univ-amu.fr (C.L.); c.dion@ims.mrc.ac.uk (C.D.); lyla.baldasseroni@univ-amu.fr (L.B.); claire.elyazidi@univ-amu.fr (C.E.-Y.); morgane.thomas@univ-amu.fr (M.T.); karine.nguyen@ap-hm.fr (K.N.); nicolas.levy@univ-amu.fr (N.L.); Jerome.robin@univ-amu.fr (J.D.R.); 2Aix-Marseille University, INSERM, INRA, C2VN, 13385 Marseille, France; richard.bachelier@inserm.fr (R.B.); alexandre.altie@univ-amu.fr (A.A.); 3APHM, Département de Génétique Médicale, Hôpital de la Timone Enfants, 13385 Marseille, France

**Keywords:** human induced pluripotent cells, differentiation, myoblasts, myotubes, motor neurons, satellite cells, Z-lines, contraction, muscular dystrophy, Facio-Scapulo-Humeral Dystrophy, Myotonic Dystrophy, Duchenne Muscular Dystrophy, Limb girdle muscular dystrophy

## Abstract

Induced pluripotent stem cells (iPSCs) obtained by reprogramming primary somatic cells have revolutionized the fields of cell biology and disease modeling. However, the number protocols for generating mature muscle fibers with sarcolemmal organization using iPSCs remain limited, and partly mimic the complexity of mature skeletal muscle. **Methods:** We used a novel combination of small molecules added in a precise sequence for the simultaneous codifferentiation of human iPSCs into skeletal muscle cells and motor neurons. **Results:** We show that the presence of both cell types reduces the production time for millimeter-long multinucleated muscle fibers with sarcolemmal organization. Muscle fiber contractions are visible in 19–21 days, and can be maintained over long period thanks to the production of innervated multinucleated mature skeletal muscle fibers with autonomous cell regeneration of PAX7-positive cells and extracellular matrix synthesis. The sequential addition of specific molecules recapitulates key steps of human peripheral neurogenesis and myogenesis. Furthermore, this organoid-like culture can be used for functional evaluation and drug screening. **Conclusion:** Our protocol, which is applicable to hiPSCs from healthy individuals, was validated in Duchenne Muscular Dystrophy, Myotonic Dystrophy, Facio-Scapulo-Humeral Dystrophy and type 2A Limb-Girdle Muscular Dystrophy, opening new paths for the exploration of muscle differentiation, disease modeling and drug discovery.

## 1. Introduction

Since the discovery in Human of somatic cell reprogramming into pluripotent stem cells (hiPSCs), protocols aimed at modeling skeletal muscle differentiation have been lagging behind those of other cell lineages [1].

In vivo, skeletal myogenesis involves tightly controlled spatial and temporal cues, and most of the current strategies take advantage of the cascade of events of somitogenesis during embryogenesis. In this context, available protocols, the vast majority of which are only focused on the production of myotubes, are either based on the induced expression of exogenous myogenic genes (*PAX3*, *PAX7*, *MYOD1*) [2,3,4] or the use of small molecules [5,6,7,8,9,10] with variable degrees of efficiency [2,11,12,13,14]. Indeed, regardless of the protocol, one of their commonalities is the duration of the differentiation process and yield. More importantly, the resulting cultures are difficult to maintain in the long term, and only partially recapitulate the organization and function of mature muscle with a lack of sarcolemmal architecture and expression of fetal or embryonic Myosins, as well as suffering from reduced functionality. Yet, in vitro systems which are able to recapitulate in vivo skeletal muscle differentiation are needed for basic research, disease modeling and drug discovery for a wide range of neuromuscular and muscular disorders for which the molecular mechanisms remain unclear or are lacking a therapy.

Using hiPSCs, we describe a human skeletal muscle differentiation model for the production of muscle fibers. Conditions of differentiation were optimized in cells from healthy donors and applied to cells from patients affected by different neuromuscular disorders. We reproducibly obtained aligned and mature contractile myofibers, together with motor neurons in conditions that preserve in the long term a pool of PAX7-positive (PAX7+ ) cells which is able to renew in culture, thereby ensuring the continuous production of muscle fibers over a long period of time. Differentiated cells are functional and respond to different drugs, opening new perspectives for the modeling of neuromuscular disorders and drug discovery.

## 2. Results

### 2.1. Concomitant Induction of Myogenesis and Neurogenesis for the Production of Innervated Muscle Fibers

During embryonic development, precise changes in the time and concentration of signaling molecules determine the characteristics of different cell lineages. Based on the use of small molecules that mimic the activation or repression of the signaling cascades required for in vivo myogenesis, but also neurogenesis, we sought to optimize conditions to produce innervated mature muscle fibers; to this end, hiPSCs were collected by mechanical dissociation. Small hiPSC clumps were plated on Matrigel-coated dishes in the presence of Thiazovivin in differentiation medium (DM) supplemented with ITS-A, LDN193189 (LDN), a potent BMP pathway inhibitor, and CHIR99021, a GSK3 inhibitor (Figure 1A). Cells were maintained for 6 days in this medium with daily medium change. On the 7th day, cells were grown in DM + LDN supplemented with Insulin Growth Factor 1 (IGF1) and Hepatocyte Growth Factor (HGF) for 24 h to favor cell lineage orientation. LDN and HGF were then removed and the cells were maintained in DM + IGF1 for four more days (until day 12), during which they proliferated. During this period (D10-12), cells could be collected by dissociation with accutase for freezing, thawing, expansion and replating. Between D6 and 12, we observed populations of cells with different morphologies that migrated out of the different cell clumps. Modulation of the BMP and Wnt pathways induce the production of neural stem cells, and LDN induce both muscle and neural stem cell differentiation, suggesting the presence of mesodermal progenitors and neuronal cells in this time window. To assess whether neuronal differentiation might be activated in parallel to muscle differentiation, we tested the effect of the transient addition of DAPT, a γ Secretase and Notch pathway inhibitor that induces cell cycle exit and postmitotic motor neuron differentiation [15]. By bright field imaging at D17 and after the addition of DAPT, we observed axons together with large patches of elongated contractile fibers (Figure 1A) with the first fibers contractions visible 2 to 4 days after the removal of DAPT (Days 19–20, Appendix A). In the absence of DAPT, rare contractions of isolated fibers were visible approximately 2 months postdifferentiation, indicating that the activation of neurulation and differentiation of motor neurons accelerate muscle cell differentiation.

### 2.2. Modulation of Wnt, BMP and Notch Pathways Leads to the Progressive Enrichment in Skeletal Muscle Progenitors, Followed by the Formation of Myofibers with Complete Sarcomeric Differentiation and Highly Organized Myofibrillar Pattern

At the different steps of our procedure, we analyzed the expression of *PAX3* and *PAX7* that characterize muscle and neuronal progenitors. Using flow cytometry, we detected approximately 12% of PAX3-positive (PAX3+ ) cells and 2% PAX7+ cells at D8 and 2–5% of PAX7+ cells from D21 onwards. By immunostaining, PAX3 was detected as early as D6, and remained visible at D8 (Figure 1B). At D12, cells expressed *PAX7*, indicating the presence of progenitors. Upon medium change and addition of DAPT, we observed a progressive change in cell shape and the appearance of elongated cells (Figure 1A), positive for the sarcomeric Desmin marker (DES, Figure 1B), with a progressive enrichment and spontaneous contraction from Day 19 onward (Appendix A).

At Day 30, multinucleated elongated muscle fibers were visualized using anti-Titin, a giant protein that links the Z disk and M lines of the sarcomere, and anti-Desmin (DES), which is responsible for connecting myofibrils to each other and to the plasma membrane (Figure 2A). As further evidenced by transmission electron microscopy (TEM), the myofibers displayed a well-organized sarcomeric structure with clearly visible A-band of Myosin filaments, I-bands of Actin filaments, and M- and Z-lines, all hallmarks of mature sarcomeres (Figure 2B). The myofibers showed multiple nuclei at the periphery, a marker of differentiation and maturation (Figure 2B), as well as large mitochondria with well-organized cristae, indicative of a high metabolic activity and consistent with the continuous contraction of the myofibers.

### 2.3. Myotubes and Motor Neurons in Close Proximity Connect with Each Other to Form Functional Neuromuscular Junctions

To address whether contractions observed as early as D19-21 occurred spontaneously or were associated with nervous stimuli enhanced by the addition of DAPT, the presence of acetylcholine receptors (AChR) and neuromuscular junction (NMJ) were visualized by staining with Alexa Fluor 555 α Bungarotoxin (Thermo Fisher Scientific, Eugene, OR, USA: BTX, white, Figure 2C) at the surface of myofibers with axons (NF staining, green) that extended toward these α Bungarotoxin clusters. Fiber contractions were irreversibly blocked by the addition of α Bungarotoxin (responsible for the inhibition of acetylcholine receptors) and Tetrododotxin (TTX, a sodium channel blocker that irreversibly inhibits action potential) to the medium (Figure 2D), advocating for the presence of motor neurons, NMJs and undifferentiated acetylcholine receptors clusters at the surface of the muscle fibers.

### 2.4. The Activation of Genes Required for Muscle Contraction Occurred Between D17 and D30

To follow the gene expression profiles at key steps of the differentiation process, we performed an RNA Seq transcriptome analysis in hiPSC clones derived from two different control cells at D8, D17 and D30, corresponding to the main medium changes (Figure 1A). As illustrated by volcano plots, the number of differentially expressed genes (DEG) was high between D8 and D17, and decreased between D17 and D30 (Appendix A).

By selecting genes with a Log2 fold change (Log2FC) of 2 and an adjusted *p* value (*p*adj) < 0.05, and comparing DEGs between D8–D17 and D17–D30 corresponding to GO terms related to “contraction”, we obtained a first list of 101 genes subdivided into four groups (Figure 2E). The first group (13 genes), activated at D8 and silenced at D17–D30, corresponded to genes involved in “myocytes adhesion and cell communication”, “muscle system process” or “cardiac conduction”. The second group of 18 genes, transiently activated at D17 and silenced at D30, corresponded to genes involved in smooth muscle and cardiac muscle contraction. The third group (18 genes), activated at D17 and decreased at D30, corresponded to genes involved in “calcium ion transmembrane transport via high voltage-gated calcium channel”, and “cardiac action potential and membrane depolarization”, suggesting—as observed in group 2—the transient activation of genes involved in cardiac contraction. Consistent with the first contractions at D19–21, the fourth group (52 genes), specifically activated at D30, corresponded to “muscle filament sliding” (21 genes, *p*adj: 7.75e^−42^), “relaxation of skeletal muscle”, “detection of muscle stretch” and “regulation of twitch skeletal muscle contraction”, indicating a progressive enrichment in pathways corresponding to skeletal muscle function. To functionally characterize hiPSC-derived myofiber and contractile activity, we analyzed intracellular calcium (Ca^2+^) release by adding Fluor8 to the fresh medium 30 min prior to recording (Figure 2F). As illustrated (Appendix A), recording easily permits separate analysis of neuronal activity (visible as small green dots) from calcium trafficking in muscle fibers. The analysis of muscle spontaneous Ca^2+^ transients per minute and per fiber at Day 30 postdifferentiation, but also in the long term, did not reveal any difference in calcium release between early and late time points. This indicated that the hiPSC-derived muscle fibers were functional, and that muscle functionality was maintained, even for long periods, in culture. 

### 2.5. Transcriptomic Profiling Showed a Two-Phase Process with Induction of Neuronal Differentiation followed by Induction of the Myogenic Program

A global analysis of DEGs between the three RNA Seq datasets (D8, D17, D30) revealed a list of 2229 DEGs between D8 and D17, and 281 genes between D17 and D30, with an overlap of 302 genes common to the three time points, i.e., corresponding to genes induced at an early step of differentiation and maintained over time (Figure 3A). For this group, the fifteen most significant associated GO terms corresponded to skeletal muscle function and development or axonogenesis (Figure 3B). Then, an analysis of significantly overrepresented biological pathways between D8–D17 and D17–D30, revealed two phases in the differentiation process. Between D8–D17, enriched pathways corresponded to axon development and synapse formation, but also cell cycle regulation, consistent with cell cycle exit and final differentiation (Figure 3C). In the second phase (D17–D30), we observed an enrichment in motor neuron networks (Figure 3D) with a marked increase in *TUBB3,* a class-III β-Tubulin restricted to neurons expressed in dorsal root ganglion and required for axon outgrowth [16] (Figure 3D,E).

### 2.6. Myogenic and Neuronal Specification and Long-Term Persistence of PAX7+ Cells

Between D17 and D30, notch pathway inhibition and activation of neuronal differentiation facilitated skeletal muscle differentiation, as evidenced by progressive activation of myogenesis and overrepresentation of genes specific to skeletal muscle differentiation and development (Figure 3F).

At the individual gene level, the expression of embryonic/fetal Myosins was followed by the progressive increase of adult isoforms (MYH2 and MYH7). We also noticed expression of *NCAM1* encoding the CD56 cell surface marker between D8 and D17 (Log2FC: -3.92, *p*adj: 3.75e^−8^) and a steady expression between D17 and D30 (Log2FC: 1.077), indicating a stable proportion of precursor cells over time. Besides *NCAM1*, the expression of other myogenic progenitors markers such as CD29 (*ITGB1*), CD82 (*TSPAN1*) and CXCR4 was also detectable in our RNA Seq datasets during the differentiation process. From D8 to D17, we observed increased *MEOX1* expression, activated in presomitic mesoderm and required for sclerotome and somitic development (Log2FC: 3.56, *p*adj: 8.49e^−6^), and an increase in *PAX3* (Log2FC: 3.03, *p*adj: 6.93e^−14,^ Fig. 3g). *PAX7* expression significantly increased between D17 and D30 (Log2FC: -2.46, *p*adj: 4.56e^−8^, Figure 3G), consistent with immunostaining data (Figure 1B) and replacement of PAX3+ muscle/neuronal precursors to PAX7+ muscle progenitors. Regarding myogenic markers (Figure 3G), *MYF5* was expressed at a low level at the different time points, while *MYOD1* strongly increased (Log2FC: -3.56, *p*adj: 3.42e^−5^) between D17–D30. Late differentiation markers, i.e., *MYOG* and *MYF6*, were poorly expressed between D8 and D17, but progressively increased between D17 and D30 (Log2FC: -2.89, *p*adj: 5.27e^−6^; Log2FC: −1.92, *p*adj: 0.08, respectively), consistent with their role in the late differentiation stages and myotube/myofiber formation.

DEGs for muscle-associated GO terms (Figure 3H) provided a list of 21 genes expressed in other mesoderm-derived tissues between D8 and D17 whose expression decreased between D17 and D30. Between D17–D30, we obtained a list of 111 upregulated genes that encode proteins involved in skeletal muscle function. This list includes muscle-specific Myosin Heavy Chains (MHC) such as embryonic/fetal *MYH8*, *MYH3*, but also adult *MYH7* and *MYH2* isoforms, as well as Troponin T such as *TNNC2*, *TNNT3* or *TNNT2*, expressed in fast type skeletal muscles, and *TNNC1*, expressed in slow type skeletal muscle (Figure 3H). We also detected increased expression of two recently discovered genes that encode the muscle-specific membrane proteins required for fusion and myotube formation, *MYMK* (Myomaker) and *MYMX* (encoding Minion, Myomerger or Myomixer) [17,18]. Myomaker required for membrane hemifusion is activated in the early differentiation stages (Log2FC: −9.84; *p*adj: 7.63E^−10^ between D8 and D30) while Myomixer required for fusion completion is activated at a later stage (Log2FC: −3.7; *p*adj: 8.42E^−7^ between D17 and D30), consistent with its action in fusion of Myomaker-positive cells [17] and the progressive increase in muscle fibers in our experimental conditions.

In addition, overrepresentation tests revealed a strong enrichment in the gene encoding components of the extracellular matrix (ECM) between D17 and D30, (Figure 3F, Appendix A), in which we noticed different Laminins, members of the collagen family (Appendix A, *p*adj: 1.26e^−10^), and activation of the TGFβ pathway. The synthesis of ECM by hiPSC-derived cells likely contributes to the organization of the innervated muscle tissue, but also to its long-term maintenance and mechanical properties without the massive cell detachment and cell death that usually occurs for primary myoblasts or hiPSC-derived myotubes after the first contractions [19].

### 2.7. Months-Long Persistence of Muscle and Motoneurons Signature

For a selected number of genes, RNA seq data (Appendix A) were confirmed by RT-qPCR from D6 to D30 and at late time points (3 months, 5 months, 7 months) postdifferentiation (Appendix A). The expression of the adult *MYH2* isoform was detectable as early as D8, and remained steady with time (Appendix A). The same kinetics were observed for Desmin (*DES*), Titin (*TTN*), Sarcoglycan Gamma Sarcolemmal protein (*SGCG*), *RYR1* encoding the ryanodine receptor involved in calcium release in the sarcoplasmic reticulum (Appendix A) and myogenic transcription factors (*MYOD1*, *MYF5*, *MYOG*, MYF6, Appendix A). The expression of *PAX3* was stable from D6–D21, and then decreased at D30, while the expression of *PAX7* progressively increased and remained stable over time, suggesting that precursor cells which are able to regenerate the culture persist over time (Appendix A). The kinetics of *PAX7*, *Desmin* and *MYH2*, *3* and *8* expression were confirmed by western blotting (Appendix A), all of which remained detectable for up to 7 months postdifferentiation (Appendix A) with a decreased level of PAX7 protein level, consistent with the high rate of differentiation and the persistence of a minority population of precursor cells in our culture.

The expression of motor neuron-specific markers such as the HB9 homeobox protein (*MNX1*) was higher in the early differentiation stages and decreased at D30, while *ISLET1* (*ISL LIM homeobox 1*) expression, which increased upon DAPT induction, remained stable. Consistent with fiber contractions, the presence of NMJs and the inhibitory action of αBTX, the expression of Choline-O-Acetyltransferase (*ChAT*) specific to cholinergic neurons and Agrin (*AGRN*) increased from D17 onwards and remained stable (Appendix A). Together with the expression of the Myelin Basic Protein (MBP, Appendix A), we also detected cells positive for S100β that colocalized with αBTX-stained NMJs, indicating the possible presence of Schwann cells (Figure 4A) and neural crest progenitors, as evidenced by increased expression of *SOX2* or *TFAP1* between D8 and D17, followed by a stable expression level between D17 and D30 (Appendix A). In agreement with the formation of functional NMJs, TEM revealed basal lamina invaginations and the presence of synaptic cleft at the surface of muscle (Figure 4B). Thus, the self-organization between muscle cells and motors neurons yielded highly organized innervated muscle fibers with NMJs at the surface of the fibers.

### 2.8. Response to Pharmaceutical Drugs

To evaluate the potential application of hiPSC-derived innervated myofibers as a preclinical model for drug testing, we tested the response to different classes of pharmaceutical agents by real-time video microscopy and calcium handling analysis. Cells at D45 were treated for different durations with different concentrations of each molecule (Figure 4C–F). For glutamate, we did not observe any difference in calcium release after the addition of 2 μM of the molecule, but increased calcium release activity was observed 20 to 30 min for a 20 μM concentration, consistent with the expression of glutamate receptors and the participation of glutamate in modulating acetylcholine transmission (Figure 4C).

We then tested the action of three muscle relaxants: Salbutamol, a β2 adreno receptor agonist with a potent relaxant property (Figure 4D), Carisoprodol, that modulates γ-Aminobutyric acid type A receptor (GABA_A_Rs) postsynaptic neurotransmission (Figure 4E) and Mexiletine, a sodium channel blocker (Na_v_ 1.4) used to reduce muscle stiffness, tiredness and weakness, in particular in DM [20] (Figure 4F).

Salbutamol decreased Ca^2+^ handling as early as 4 h (h) after 1 μM and 10 μM drug addition to the medium, with recovery to basal activity after 24 h (Figure 4D). Carisoprodol modulated postsynaptic neurotransmission and reduced muscle contraction 3 to 4 h after the addition of 0.5 to 1 mM drug, with total recovery after 24 h (Figure 4E). As expected, given its ability to block the sodium channel and inhibit Na ^+^/Ca^2+^ exchange-dependent Ca^2+^ overload, we observed a decrease in calcium handling 2 h after the addition of 10 μM mexiletine, and a sharp decrease 1 to 2 h after 100 μM of the drug was added, followed by a progressive but partial recovery between 3 to 24 h posttreatment (Figure 4F). Altogether, the functional responses to all these different drugs highlight the value of our “innervated muscle in a dish” model for drug screening and future therapeutic development.

### 2.9. HiPSC-Derived Muscle Cells for Modeling Neuromuscular Disorders

To evaluate the reproducibility and robustness of our protocol, we applied it to the four most frequent muscular dystrophies, i.e., Duchenne Muscular Dystrophy (DMD, OMIM 310200), Myotonic Dystrophy (DM1, OMIM 160900), Facio Scapulo Humeral Dystrophy (FSHD2, OMIM 158901) and Limb Girdle Muscular Dystrophy 2A (LGMD2A, or LGMDR1, OMIM 253600). Using antibodies against Desmin and Titin, we observed the formation of multinucleated, millimeter-long, aligned myofibers and a mature fiber striated pattern at D30 postdifferentiation (Figure 5A). We also observed immature AChR clusters at the surface of the fibers after α BTX staining (Figure 5B). Compared to our two control cells and the two clones analyzed for DM1 and LGMD2A, we observed that the fiber section size was significantly reduced in DMD cells, and wasvariable as well as slightly increased in FSHD2 cells (Figure 5C).

Furthermore, for the four diseases, TEM revealed the full maturation of hiPSC-derived muscle fibers with visible Z lines structures (Figure 6A). However, we observed an increased proportion of discontinuous and tortuous architectures with interrupted Z lines, suggesting defects in sarcolemmal anchoring, as reported in [21,22]. In DMD, the presence of vacuole-like areas between the fibers (Figure 6A) strikingly resembled what was reported in [23] in vivo as a sign of muscle fiber degeneration. In LGMD2A, we observed a patchy pattern of small disorganized fibers with electron-dense inclusion between fibers, as reported in [24,25].

Having shown that the codifferentiation of neurons and myotubes gives rise to the formation of NMJs at the surface of the fibers in healthy and diseased contexts, we next investigated muscle fiber functionality by measuring the calcium transient at D30. For all diseases, we observed a significant decrease in the calcium handling amplitude in DMD cells (*p* < 0.0001) compared to controls, but an increase in other pathologies (DM1, FSHD and LGMD; *p* < 0.0001, Figure 6B).

The expression of the different genes validated in control cells was evaluated in these different clones by RT-qPCR from D6 to D30 (Appendix A, Figure 5D). The expression of embryonic and fetal MyHCs was comparable in the different contexts, while that of the adult isoform (*MYH2*) remained very low, in particular in DMD, and was more variable between diseases, suggesting delayed muscle maturation in these different clones (Appendix A). The expression of *Desmin*, *Titin* and *PAX7* was comparable between conditions.

Finally, to address whether the approach described here might be applied to modeling the large repertoire of inherited muscle diseases, we classified our RNA-seq data based on the list of genes used for diagnosis by whole exome sequencing of gene panels for the different classes of diseases (Appendix A). This preliminary analysis revealed that the vast majority of these genes were expressed in our experimental conditions. Thus, together with the visualization of structural abnormalities, these analyses confirmed that our protocol can be reliably used to model neuromuscular pathologies including at the functional level, providing a reliable cellular model and potential readout for functional testing. The analyses further highlight the value of our model to reveal signatures associated with genetic neuromuscular diseases.

## 3. Discussion

Different protocols exist for the differentiation of human pluripotent or embryonic stem cells for a large variety of terminally differentiated tissues. Nevertheless, differentiation of the skeletal muscle lineage remains challenging [5,26,27,28,29], with some methodologies relying on the ectopic expression of muscle-specific transcription factors by viral gene delivery [12,28,29,30,31] and limited enrichment in mature skeletal muscle fibers [2,12,13,14].

Based on the selective and sequential addition of small molecules, we describe a novel and reliable procedure for the production of innervated and functional skeletal muscle tissue. Our serum-free protocol stems from the dual modulation of the Wnt and BMP pathways that induce paraxial development [7,8], but incorporates an additional step of inhibition of the notch pathway, which increases myogenic specification, and facilitates the production of motor neurons and muscle maturation [15,32,33].

We showed that our system recapitulates key functional aspects of human skeletal muscle organization, including in its ultrastructural architecture, development of a functioning contractile apparatus and capacity for long-term regeneration that allowed culture for up to a year. Compared to other reports in which gradual attenuation and exhaustion were observed [7,34], the presence of hiPSC-derived motor neurons that codifferentiate with muscle fibers accelerates the differentiation.

A critical bottleneck to differentiation protocols published so far is the maturation and functionality of hiPSC-derived muscle fibers. In controls cells, we observed intact sarcomeres with distinct I-bands or M- and Z-lines without any aberrant myoblast formation with branched arms as reported [7,32]. Furthermore, if sarcomeric organization can be observed in some fibers with other approaches [7,35], hiPSC-derived muscle fibers are mostly composed of embryonic or fetal Myosins [7,35,36]. We showed here that fetal/perinatal, but also adult, Myosin heavy and light chains are detectable at the time of the first contractions, with a stable level of expression overtime.

Furthermore, most protocols for the production of innervated muscle fibers rely either on the coculture of human primary myoblasts with animal motor neurons extracted from dorsal roots ganglions, the coculture of hiPSC-derived motor neurons with human or animal muscle cells [37] or, more recently, hiPSC-derived motor neurons mixed with hiPSC-derived muscles cells [38].

We report here on the simultaneous codifferentiation between both cell types in healthy and diseased backgrounds where hiPSC-derived motor neurons and muscle cells self-organize and establish functional contacts with muscle fibers in 2D. The presence of acetylcholine receptors at the surface of the fibers for the formation of a contractile “muscle in a dish” further confirm the important contribution of the different lineages forming muscle, as observed in vivo [38,39,40,41]. Albeit based on the RNA seq data, the fetal acetylcholine receptor subunit isoform was more abundant than the adult isoform at D30 postdifferentiation. A recent article reported the production of human trunk neuromuscular organoids in which maturity was reached at day 50 postdifferentiation [42]. Interestingly, using a different approach, we reached the same conclusion, i.e., that the simultaneous differentiation of muscle and peripheral neurons is necessary for the production of mature and functional muscle fibers, an important prerequisite for disease modeling.

Our system enabling codifferentiation between skeletal muscle cells and motor neurons from the same individual will deepen our understanding of the early stages of muscle-motor neuron interactions and NMJ formation in physiological and pathological contexts, and open new perspectives for the investigation of early differentiation steps or muscle-nerve interactions. 

Culture on matrigel coating that contains laminin, a component of the basal lamina of NMJs, Collagen IV and Proteoglycans [43], is highly favorable for motor neuron differentiation, the maintenance of multinucleated muscle fibers, but also PAX3/PAX7 positive cells, over a long period of time. The gradual activation of genes encoding ECM factors, including laminins, collagens and proteoglycan, suggests a progressive replacement of matrigel by this ECM, which likely contributes to long-term cell maintenance, functionality and attachment, despite continuous contractions usually leading to muscle detachment. 

Overall, the lack of successful strategies for the production of functional and fully mature myofibers jeopardizes investigations into disease mechanisms and therapeutic development. Furthermore, as genome-wide approaches often prove difficult regarding the interpretation of variants of unknown significance (VUS), this protocol provides a novel tool for the evaluation and deciphering of pathways associated with diseases causing mutations and associated signatures. Thus, compared to the occasional twitching reported for control cells [7,32], or the absence of twitching in patient cells [32], our straightforward procedure fills the gap between currently available 2D protocols, in which maturity and functionality are reduced, and the advances provided by neuromuscular organoids [42]. Compared to organoids, contractions were visible as early as 19–21 days postdifferentiation in our conditions. This reliable and easy-to-use model provides a new tool to investigate the early stages of differentiation and cell–cell interactions/interplay, and makes it possible to model a wide range of neuromuscular disorders, thereby facilitating drug discovery.

## Figures and Tables

**Figure 1 cells-09-01531-f001:**
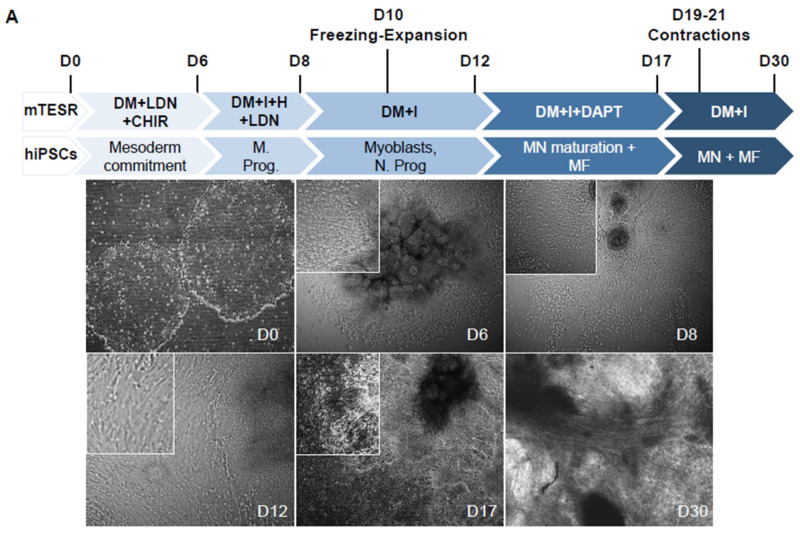
Protocol for myogenic differentiation. (**A**) Time line and phase contrast images of cells differentiated from human induced pluripotent stem cells (hiPSCs) at Day 0 (D0), D6, D8, D12, D17 and D30, corresponding to changes in the composition of the medium. Thiazovivin, an inhibitor of the ROCK pathway, was added at the time of plating and two days after. The composition of the medium is indicated for the different culture steps. CHIR: CHIR99021; LDN: LDN193189; F: FGF2; I: IGF1; H: HGF; DAPT: N-[N-(3,5-Difluorophenacetyl)-L-alanyl]-S-phenylglycine t-butyl ester. Cells could be frozen between Days 10 and 12 for storage, expansion and further differentiation. First contractions were visible between day 19 and 21 postdifferentiation. (**B**) Representative Z stack after confocal microscopy of cultures stained with anti-PAX3 antibody at Day 6 and Day 8 postdifferentiation, and Day 12 with anti-PAX7. Representative immunostaining of early myofibers at Day 14 stained for the sarcomeric Desmin (green). Cells were counterstained with DAPI.

**Figure 2 cells-09-01531-f002:**
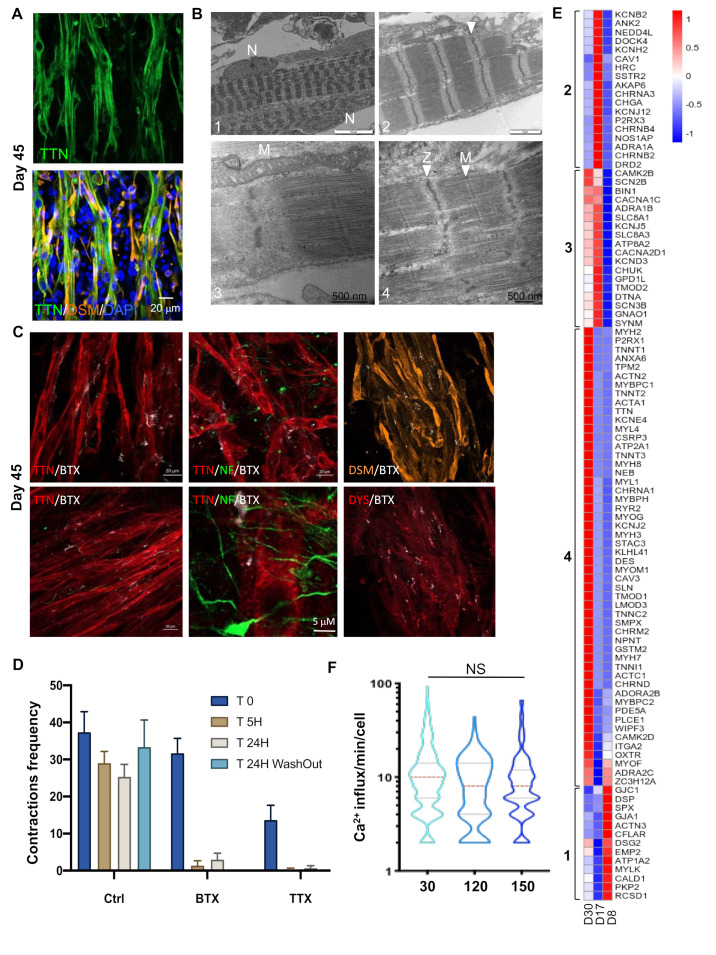
Ultrastructural and functional validation of hiPSC-derived myofibers. (**A**) Representative Z stacks of muscle fibers at Day 30, stained for Titin (TTN, green, upper panel) and merged between Desmin (DES, orange) and Titin filament (green, lower panel) at Day 30 postdifferentiation. Nuclei were counterstained with DAPI. (**B**) Ultrastructural features of differentiation. Control cells showed complete sarcomeric organization with highly organized myofibrillar patterns and fully mature sarcomeric banding pattern with clearly visible Z-lines (arrows, panels 2; 4), nuclei localized at the periphery of the fibers (panel 1, N) and large mitochondria (panel 3, M). Thick filaments were assembled with the formation of Z and M lines across the Myosin filaments (Panel 4). Z-lines were aligned, forming a clearly visible structural pattern (panels 1;2;4). We also observed I-bands of Actin filaments and A-bands (panels 2; 4). (**C**) Representative Z stacks of cultures stained for acetylcholine receptors (AChR) using Alpha Bungarotoxin coupled with Alexa 555 (White), and immunostaining of muscle fibers using anti-Titin (TTN, red), anti-Dystrophin (DYS, red), anti-Desmin (orange) and anti-NeuroFilament (NF, green) antibodies. We observed several undifferentiated acetylcholine receptors clusters at the surface of the myofibers (white). (**D**) Contractions of muscle fibers were recorded in real time by video microscopy in control cells (Appendix A) prior to or after the addition of Alpha Bungarotoxin (α BTX) or Tetrodotoxin (TTX), to the cell medium, or after medium replacement and drug removal. Histogram displays the mean number of contractions ± S.D. shown by error bars, prior to drug addition (T0), 5 h (T 5H) or 24 h after (T 24 H), or 24 h after drug removal and medium replacement (T 24H washout). The addition of both drugs irreversibly inhibited cell contraction. (**E**) Heatmap of DEGs corresponding to the “contraction” GO terms in D8 vs. D17 and D17 vs. D30. Brackets identify genes differentially expressed for the different time points. Group 1, genes activated at D17; group 2, genes activated at D17 and D30; and group 3, genes upregulated at D8. (**F**) Intracellular calcium signaling in hiPSC-derived myofibers was measured using an Axio Observed microscope at day 30, day 120 (4 months) and day 150 (5 months) postdifferentiation. For each time point, >215 muscle fibers were analyzed separately. The number of contractions per minute and per fiber is reported as violin plots. No difference was observed in the calcium influx at day 120 and 150 compared to day 30 (ANOVA multiple comparison test followed by Brown-Forsythe and Welch correction *p* = 0.264 and *p* = 0.299; respectively). hiPSC-derived myofibers by transmission electron microscopy (TEM) at Day 45 post.

**Figure 3 cells-09-01531-f003:**
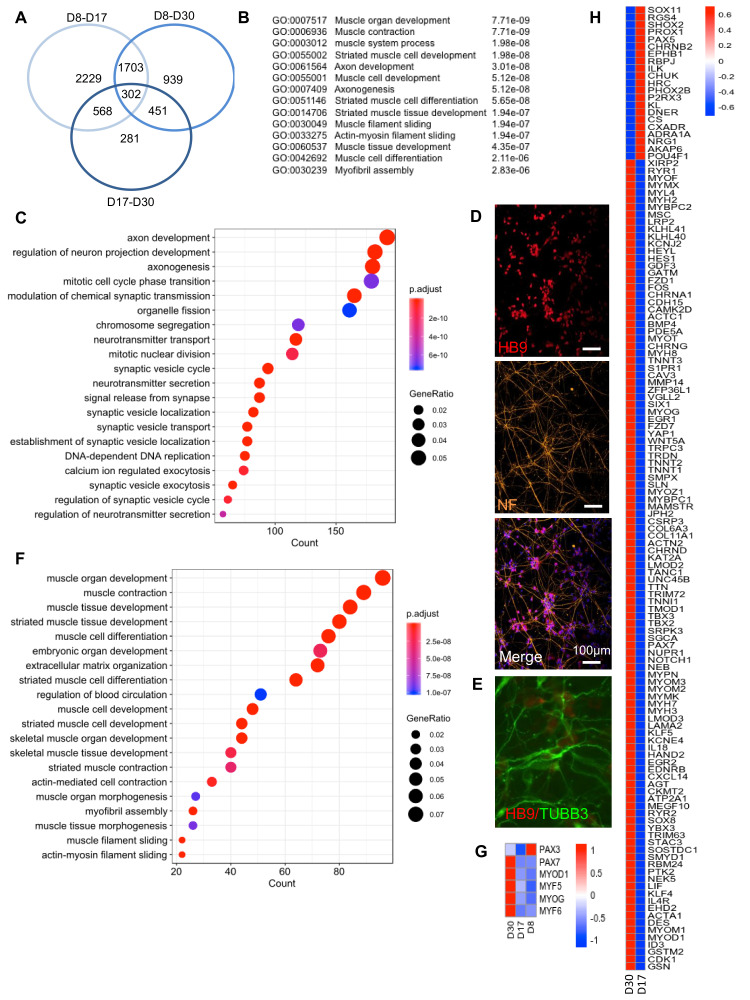
Transcriptomic profiling of in vitro differentiated innervated muscle fibers at day 8, 17 and 30 postdifferentiation. (**A**) Venn diagram showing the number of genes differentially regulated between D8 and D30, D8 and D17 or D17 and D30. The three time points correspond to the main medium changes. Genes were selected based on a *p*adj <0.05. Lists of genes that are differentially expressed between D8–D17, D17–D30 and D8–30 are given in Appendix A. (**B**) List of the most significant Biological processes (BP) corresponding to the 302 genes common to the three time points with associated GO term and *p*adj values. (**C**) Biological pathways significantly overrepresented between D8 and D17. (**D**) Representative Z stacks of hiPSC-derived motor neurons using anti HB9 antibodies (upper panel, red), anti-Neurofilament (NF, orange, medium) and merged (bottom) with nuclei counterstained with DAPI. (**E**) Immunostaining of hiPSC-derived motor neurons using anti HB9 antibodies (red) and anti-TUBβ3 (green) with nuclei counterstained with DAPI. (**F**) Biological pathways significantly overrepresented between D17 and D30. (**G**) Heatmap of the different myogenic transcription factors at D8, D17 and D30. (**H**) Heatmap of differentially expressed genes corresponding to “skeletal muscle differentiation” between D8, D17 and D30.

**Figure 4 cells-09-01531-f004:**
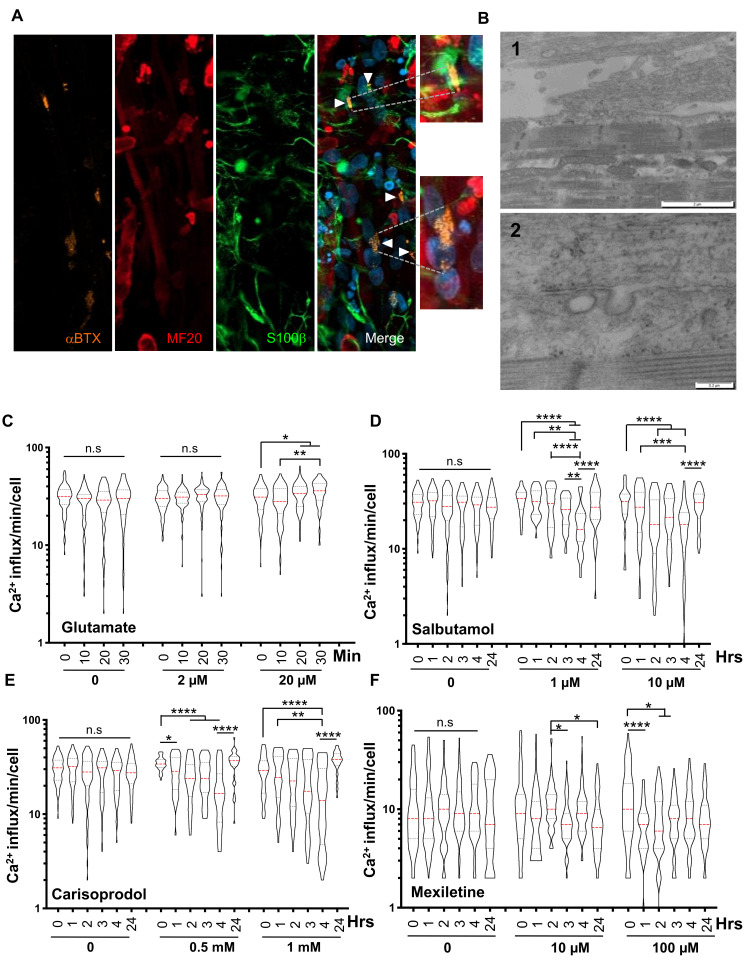
Characterization of NMJs and evaluation of the functional response to pharmaceutical drugs. (**A**) Representative Z stacks of cells stained at Day 30 postdifferentiation using alpha Bungarotoxin (aBTX, orange), anti-MyHC (MF20, red) or anti-S100β (green). NMJs are indicated by white arrows on the merge. The formation of functional NMJs was confirmed by the presence of S100β-positive cells expressed by Schwann cells capping the neuronal terminals. A higher magnification of different aBTX-positive NMJs is shown. (**B**) Transmission electron microscopy showing the presence of an axon positioned close to a muscle fiber. The basal lamina is visible (panels 1–2), as well as invagination of the plasma membrane (panel 2). (**C**–**F**) We measured the transient calcium influx in control myofibers prior to drug addition (0 h) and at different time points after a single addition of different concentrations of each drug. Violin plots display the number of influxes per minute and per fiber (*n* > 50). Statistical significance was determined using an ANOVA multiple comparison test followed by Brown-Forsythe and Welch correction and comparison to the basal condition (prior to drug addition). **C.** 0; 2 and 20 μM for glutamate after 0; 10; 20; 30 min of incubation, (**D**) 0; 1 and 10 μM for Salbutamol after 0, 1, 2, 4 and 24 h of incubation. (**E**) 0, 0.5 and 1 mM for Carisoprodol after 0, 1, 2, 4 and 24 h of incubation. **F.** 0, 10 and 100 μM for Mexiletine after 0, 1, 2, 4 and 24 h of incubation.

**Figure 5 cells-09-01531-f005:**
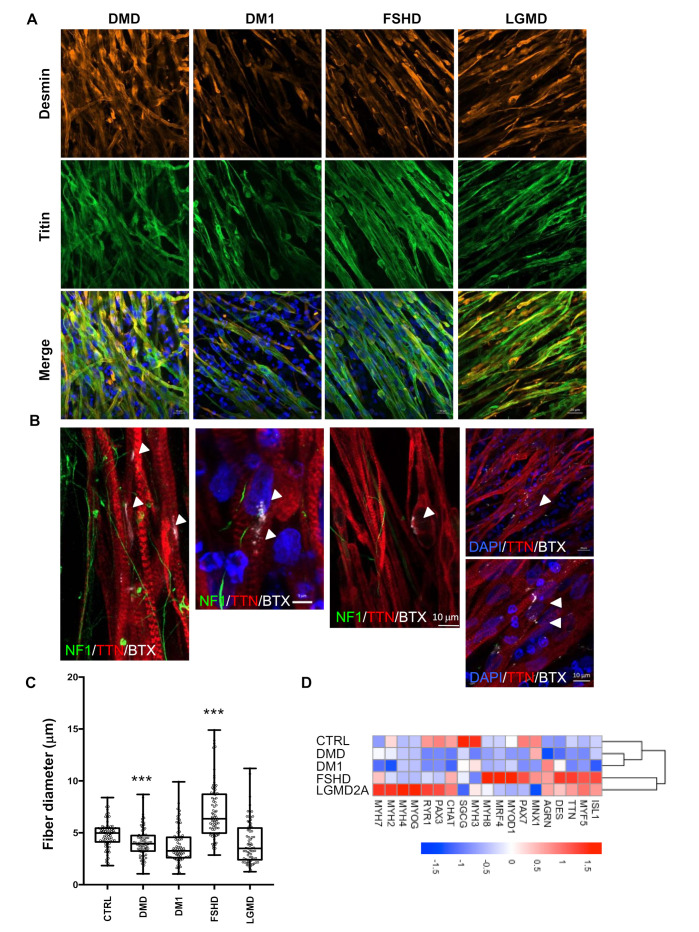
Structural and functional phenotyping of hiPSC-derived muscle cells from patients affected by muscular dystrophy. (**A**) Representative Z stacks of myofibers at day 45 with staining of sarcomeric Desmin (DES, orange) and Titin filament (TTN, green) for Duchenne Muscular Dystrophy (DMD), myotonic dystrophy (DM1), Facio Scapulo Humeral Dystrophy (FSHD) and Limb Girdle Muscular Dystrophy Type 2A (LGMD2A). (**B**) Representative Z stacks of cultures stained for acetylcholine receptors (AChR) using Alpha Bungarotoxin coupled with Alexa 555 (White, arrows) at the surface of hiPSC-derived muscle fiber stained in red using anti-Titin (TTN, red). Antineurofilament (NF, green) was used to visualize motor neurons axons. Nuclei were counterstained with DAPI. As in the control cells, we observed the presence of several undifferentiated acetylcholine receptors clusters at the surface of the myofibers for the different pathologies. (**C**) We compared the muscle fiber section of the Titin-stained fibers in control cells and the four different pathologies (DMD, DM1, FSHD or LGMD2A). For each condition, 500 fibers were analyzed, and the section of the fiber was determined at three different positions along each. Box plots display the mean sizes for the different samples. Values were compared using an ANOVA multiple comparison test, followed by Brown-Forsythe and Welch correction tests; *****p* < 0.0001; ****p* < 0.001; ***p* < 0.01. –(**D**) Heatmap for expression of different muscle and motor neuron markers over time for cells differentiated from the control (CTRL) or pathological (DMD, DM1, FSHD, LGMD2A) hiPSCs at D30 postdifferentiation, determined by RT-qPCR on biological triplicates, each realized in biological duplicates (*n* = 6 per data point). The names of the different genes are indicated on the bottom. Data for additional time points (D6, D8, D12, D17 and D21) are presented in the Appendix A section.

**Figure 6 cells-09-01531-f006:**
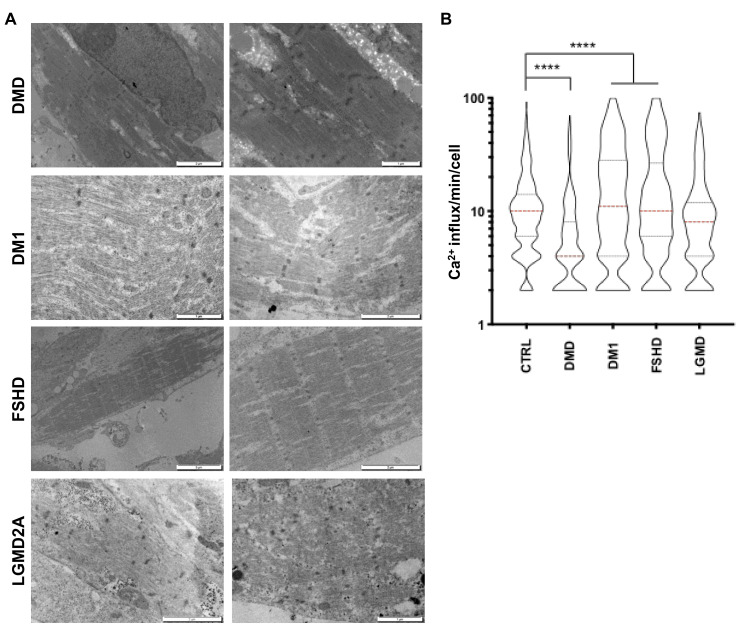
Functional and ultrastructural characterization of diseased hiPSC-derived muscle fibers. (**A**) Ultrastructural features of hiPSC-derived muscle fibers for different neuromuscular disorders. Electron microscopy was performed 60 days postdifferentiation. Representative images for DMD, DM1, FSHD and LGMD2A are presented. (**B**) Intracellular calcium signaling in hiPSC-derived myofibers was measured using an Axio Observed microscope at day 45 postdifferentiation. For each time point, >180 muscle fibers were analyzed. We report the number of influxes recorded per minute and per fiber as violin plots. Data were compared using an ANOVA multiple comparison test followed by Brown-Forsythe and Welch correction. DM1 myofibers presented lower calcium influx compared to the control (****; *p* < 0.0001), whereas calcium influx increased in FSHD and DMD myofibers (****; *p* < 0.0001). No difference was observed between the control and LGMD2A myofibers (*p =* 0.89).

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
