# Peer review of "Multilineage Differentiation for Formation of Innervated Skeletal Muscle Fibers from Healthy and Diseased Human Pluripotent Stem Cells"

_cells, 2020, doi:10.3390/cells9061531_

Round 1

Reviewer 1 Report

The manuscript from Mazaleyrat et al. claims the generation of innervated skeletal muscle fibers from healthy and diseased human pluripotent stem cells. The author showed that their system recapitulates key functional aspects of human skeletal muscle organization, development of a functioning contractile apparatus and capacity of long-term regeneration. This study was well designed supported their conclusion, but this manuscript will be needed additional data to support the authors' claims. My specific comments are below:

Specific comments:

  1. Figure 1A: In this study, it seems that the differentiation was performed according to the time line. Except for ICC, gene expressions have to check at each time point for cell identification.
  2. Figure 1B: In a recent study by Darabi group (STEM CELLS, 2011), selective induction of myogenic markers PAX3 and PAX7 were used after embryonic body formation to improve the myogenic differentiation efficiency and constitutes a source of myogenic cells of importance for skeletal muscle formation (Relaix group, Nature 2005). In muscle progenitor cells, what percentage of cells are expressing those genes?
  3. Figure 3: Is there any reason to select D8, D17 and D30 for RNA-SEQ analysis? After DAPT addition, the author observed exons together with large patches of elongated contractile fiber. Therefore, before DAPT treatment, D12 sample is also important.
  4. Figure 4: To evaluate the potential application of hiPSC-derived innervated myofibers, the author tested the response to different pharmaceutical agents of Calcium handling analysis, however, it is necessary to test contraction of the skeletal muscles, demonstrating that muscle activity was dependent upon functional NMJs. To investigate the functionality of neuromuscular junction (NMJs), Martins group (Cell stem cell, 2020) measured the contractile activity of cells before and after treatment with curare (blocker of AChRs). Also, author should use disease-derived iPSCs to compare control (healthy).
  5. Discussion: the author mentioned that “author attained the same conclusion that the simultaneous differentiation of muscle and peripheral neuron is necessary for production of mature and functional muscle fiber”. What are the advantages over Martins group (Cell stem cell, 2020)?

Author Response

We thank both reviewers for their careful reading of our manuscript and helpful comments. Our point-by-point response is provided below. All remarks have been carefully addressed.

We believe that the manuscript has greatly improved and is more suitable for publication in Cells.

With respect to reviewer 1.

The manuscript from Mazaleyrat et al. claims the generation of innervated skeletal muscle fibers from healthy and diseased human pluripotent stem cells. The author showed that their system recapitulates key functional aspects of human skeletal muscle organization, development of a functioning contractile apparatus and capacity of long-term regeneration. This study was well designed supported their conclusion, but this manuscript will be needed additional data to support the authors' claims. My specific comments are below:

Specific comments:

  1. Figure 1A: In this study, it seems that the differentiation was performed according to the time line. Except for ICC, gene expressions have to check at each time point for cell identification.

Authors’ response: Expression of a number of key genes (19 genes) has been analyzed and quantified at different time points (day 6, 8, 12, 17, 21 30 post differentiation) by RT-qPCR. The protein level of 4 different muscle markers (MyH2, MyH3, MyH8 and Desmin) has been quantified by Western blot at day 6, 8, 12, 17, 21, 30, 90, 150, 210 post differentiation. Data are presented in supplementary figure 2 for RT-qPCR assays and supplementary figure 3 for western blots. The same type of analyses has been done for the different pathologies as well, supplementary figures 4-7 for RT-qPCR and 8-9 for western blots.

  1. Figure 1B: In a recent study by Darabi group (STEM CELLS, 2011), selective induction of myogenic markers PAX3 and PAX7 were used after embryonic body formation to improve the myogenic differentiation efficiency and constitutes a source of myogenic cells of importance for skeletal muscle formation (Relaix group, Nature 2005). In muscle progenitor cells, what percentage of cells are expressing those genes?

Authors’ response: We thank reviewer 1 for this question. We have added the following sentence: “Using flow cytometry; we detected approximately 12% of PAX3-positive (PAX3+) cells and 2% PAX7+ cells at D8 and 2-5% of PAX7+ cells from D21 onwards” lines 93-95 to answer this question.

  1. Figure 3: Is there any reason to select D8, D17 and D30 for RNA-SEQ analysis? After DAPT addition, the author observed exons together with large patches of elongated contractile fiber. Therefore, before DAPT treatment, D12 sample is also important.

Authors’ response: we agree with this comment but decided to analyze cells at time points corresponding to the main changes in the medium for RNA Seq while additional confirmations (RT-qPCR, western blotting and IF) were done at intermediate time points (see response to point 1).

  1. Figure 4: To evaluate the potential application of hiPSC-derived innervated myofibers, the author tested the response to different pharmaceutical agents of Calcium handling analysis, however, it is necessary to test contraction of the skeletal muscles, demonstrating that muscle activity was dependent upon functional NMJs. To investigate the functionality of neuromuscular junction (NMJs), Martins group (Cell stem cell, 2020) measured the contractile activity of cells before and after treatment with curare (blocker of AChRs). Also, author should use disease-derived iPSCs to compare control (healthy).

Authors’ response: We thank reviewer 1 for these remarks. As presented in Figure 2D, muscle contraction has been monitored by cells live imaging in the absence of drugs or later treatment with Tetrodotoxin and alpha bungarotoxin. We have shown that both drugs are able to permanently block muscle contraction (Figure 2D).

  1. Discussion: the author mentioned that “author attained the same conclusion that the simultaneous differentiation of muscle and peripheral neuron is necessary for production of mature and functional muscle fiber”. What are the advantages over Martins group (Cell stem cell, 2020)?

Authors’ response: We believe that our work has three advantages or is complementary to Martins’s group on three aspects: the ease of realization, the possibility to obtain contractile fibers in 19-21 days compared to 50-100 days and the possibility to freeze and further expand cells at intermediate time points (Days 10-12).

This is mentioned in the discussion section, lines 442-443;even if the fetal acetylcholine receptor subunit isoform is mostly expressed at D30 post differentiation » and lines 469-472:As compared to organoids, contractions are visible as early as 19-21 days post differentiation in our conditions. This reliable and easy to use model provides a new tool to investigate early stage of differentiation, cells-cells interactions/ interplay and allows for the modeling of a wide range of neuromuscular disorders and drug discovery. . »

Reviewer 2 Report

Line 66

What do the authors mean with ‘mature hiPSC’ ? – unusual term.

Lines 74-78

The authors claim the presence of “populations of cells with different morphologies that migrated out of the different cell clumps. Modulation of BMP and Wnt pathways induce production of neural stem cells and LDN induces both muscle and neural stem cells differentiation indicating the presence of mesodermal progenitors and neuronal cells in this time window.’  Is this a claim for having neural stem/progenitor cells in the dish ? Can you picture these different populations, are you willing to characterize them further ?

Line 222-225

NCAM1 can also be a marker for myotube / myofiber generation. Here Pax7 or CD82, CXCR4, CD29 would be better markers for myogenic progenitors.

Line 229

The authors claim ‘replacement of PAX3+ muscle/neuronal precursors to PAX7+ muscle progenitors.’  Actually, Pax7 could also mark neural crest populations. To support their claim, they could co-stain for TFAP2A or SOX10, SOX2.

Line 326 ff

To support the claim of fully mature muscles, the acetylcholine receptor subunit composition could be evaluated (upregulation of e subunit or predominant for γ ?).

Author Response

We thank both reviewers for their careful reading of our manuscript and helpful comments. Our point-by-point response is provided below. All remarks have been carefully addressed.

We believe that the manuscript has greatly improved and is more suitable for publication in Cells.

With respect to Reviewer 2.

Line 66

What do the authors mean with ‘mature hiPSC’ ? – unusual term.

Lines 74-78

The authors claim the presence of “populations of cells with different morphologies that migrated out of the different cell clumps. Modulation of BMP and Wnt pathways induce production of neural stem cells and LDN induces both muscle and neural stem cells differentiation indicating the presence of mesodermal progenitors and neuronal cells in this time window.’  Is this a claim for having neural stem/progenitor cells in the dish ? Can you picture these different populations, are you willing to characterize them further ?

Authors’ response: This claim is based on our review of the literature (see paragraph 2.1) on the role for the different molecules in differentiation and might indeed explain the presence of muscle and neuronal progenitors in the dish. It might be indeed interesting to further characterize these cells, for cell population sorting application for instance. However, at this step, we think that this is beyond the scope of the present study.

Line 222-225

NCAM1 can also be a marker for myotube / myofiber generation. Here Pax7 or CD82, CXCR4, CD29 would be better markers for myogenic progenitors.

Authors’ response: We agree with this comment and NCAM1 was only mentioned as it is commonly used for muscle progenitors purification and cell sorting. We have analyzed the expression of PAX7 at different time points during the culture as it is presented in the supplementary information (supplementary figure 2 for RT-qPCR assays and supplementary figure 3 for western blots; respectively). We have added a sentence (lines 228-230), indicating: “Besides NCAM1, expression of other myogenic progenitors markers such as CD29 (ITGB1) CD82 (TSPAN1); CXCR4 were also detectable in our RNA Seq datasets during the differentiation process.”

Line 229

The authors claim ‘replacement of PAX3+ muscle/neuronal precursors to PAX7+ muscle progenitors.’  Actually, Pax7 could also mark neural crest populations. To support their claim, they could co-stain for TFAP2A or SOX10, SOX2.

Authors’ response: We thank reviewer 2 for this relevant remark. As mentioned in the text and presented in figure 4A, neural crest cells are also present during the differentiation process. This is now discussed page 286-287: “…and neural crest progenitors as evidenced by increased expression of SOX2 or TFAP1 between D8 and D17 followed by a stable expression level between D17 and D30”.”

Line 326 ff

To support the claim of fully mature muscles, the acetylcholine receptor subunit composition could be evaluated (upregulation of e subunit or predominant for γ ?).

Authors’ response: we thank reviewer 2 for raising this point. We agree that additional staining would be interesting for further characterization of NMJs in controls and pathologies. Unfortunately, immuno-staining is not be feasible in the time allotted for the review of our manuscript. Nevertheless, to address this remark the following sentence has been added lines 441-445 in the Discussion section: “The presence of acetylcholine receptor at the surface of the fibers for formation of a contractile “muscle in a dish” further confirm the important contribution of the different lineages forming muscle in its maturation as observed in vivo [38-41], even if based on the RNA seq data, the fetal acetylcholine receptor subunit isoform is more abundant than the adult isoform at D30 post differentiation.

Round 2

Reviewer 1 Report

The rebuttal is reasonable and well described. Well done.

Author Response

We thank the reviewer for their positive response.